# Using Inertial Measurement Units and Machine Learning to Classify Body Positions of Adults in a Hospital Bed

**DOI:** 10.3390/s25020499

**Published:** 2025-01-16

**Authors:** Eliza Becker, Siavash Khaksar, Harry Booker, Kylie Hill, Yifei Ren, Tele Tan, Carol Watson, Ethan Wordsworth, Meg Harrold

**Affiliations:** 1Curtin School of Allied Health, Curtin University, Perth 6102, Australia; k.hill@curtin.edu.au (K.H.); m.harrold@curtin.edu.au (M.H.); 2Virtual Care and Community Care, East Metropolitan Health Service, Perth 6000, Australia; 3School of Electrical Engineering, Computing and Mathematical Sciences, Curtin University, Perth 6102, Australia; siavash.khaksar@curtin.edu.au (S.K.); yifei.ren@curtin.edu.au (Y.R.); t.tan@curtin.edu.au (T.T.); ethan.wordsworth@gmail.com (E.W.); 4Physiotherapy Department, Royal Perth Hospital, Perth 6000, Australia

**Keywords:** inertial measurement units, healthcare, monitoring, machine learning

## Abstract

In hospitals, timely interventions can prevent avoidable clinical deterioration. Early recognition of deterioration is vital to stopping further decline. Measuring the way patients position themselves in bed and change their positions may signal when further assessment is necessary. While inertial measurement units (IMUs) have been used in health research, their use inside hospitals has been limited. This study explores the use of IMUs with machine learning to continuously capture, classify and visualise patient positions in hospital beds. The participants attended a data collection session in a simulated hospital bedspace and were asked to adopt nine positions. Movement data were captured using five IMU Xsens DOTs attached to the forehead, wrists and ankles. Support Vector Machine (SVM) and K-Nearest Neighbours classifiers were trained using five different combinations of sensors (e.g., right wrist only, right and left wrist) to determine body positions. Data from 30 participants were analysed. The highest accuracy (87.7%) was achieved by SVM using forehead and wrist sensors. Adding data from ankle sensors reduced the accuracy. To preserve patient privacy in a hospital setting, a 3D visualisation was developed in Unity, offering a non-identifiable representation of patient positions. This system could help clinicians monitor changes in position which may signal clinical deterioration.

## 1. Introduction

Deterioration in health and poor outcomes in adults who are hospitalised are not uncommon [1,2,3]. In some situations, deterioration is unavoidable and even acceptable (e.g., terminal care). However, in many situations, poor outcomes could be avoided if healthcare practitioners detect untoward changes in vital signs and initiate medical treatment [4,5,6]. The earlier deterioration is recognised, the earlier treatment can be initiated, which reduces the likelihood of a poor outcome. In addition to the benefits to the person and their family, early detection of untoward changes is likely to have a substantial economic impact. This is because ‘avoidable’ deteriorations often prolong hospital stay and increase healthcare utilisation. Notably, avoidable deteriorations in health have been estimated to cost AUD 14,000 per episode [7] and account for 8% of annual health expenditure [8].

In the hospital setting, continuous monitoring with a patient-to-nurse ratio of 1:1 is reserved for those individuals who are at the greatest risk of rapid, life-threatening deterioration, such as those admitted to the resuscitation bays in an emergency department or an intensive care unit. The cost of this strategy restricts its usage beyond people who are the most unwell/unstable. In hospital wards that admit patients at high risk of deterioration (e.g., following major cardiothoracic surgery), patient-to-nurse ratios may be higher (e.g., 4:1), and a continuous real-time telemonitoring system of vital signs is used to alert healthcare practitioners to any changes in basic physiology that are outside an acceptable range. However, to date, the vital signs that can be continuously monitored are largely limited to variables collected via electrocardiograms (heart rate and rhythm) and pulse oximetry (peripheral oxygen saturation).

Currently, continuous real-time telemonitoring does not extend to include body position or bodily movement. However, the capacity to incorporate these measures may assist in the detection of avoidable deterioration. Examples include delirium, which can manifest as a reduction in movement (hypoactive delirium) or restlessness (hyperactive delirium), focal seizures or limb paralysis due to localised neurovascular events. In addition to detecting acute deteriorations, the capacity to measure and alert staff to insufficient bodily movement will assist in avoiding the known sequelae of prolonged recumbency, such as pressure ulcers.

Outside the hospital setting, multiple methods have been developed that allow continuous monitoring of bodily position and movement, such as the use of optical motion capture technologies, fiber-optic sensors, E-textile sensors, load cells and inertial measurement units (IMUs). Of these, IMUs offer the most promise for use in a hospital environment as they are small, unintrusive, wearable devices and can be re-used between patients. Optical motion capture technologies such as Vicon (https://www.vicon.com/ accessed on 22 August 2024) are not suitable in a hospital environment due to their high cost (approximately AUD 250,000) [9] and the need for specialist laboratory settings. Fiber-optic and E-textile sensors have seen advancements in usability [10] but may provide limited precision in detecting changes in position and have not been validated in a clinical environment. Load cells and pressure sensors have shown promise as methods for classifying the positions of people lying in bed, but are unable to classify movement when a body part is not in contact with a bed [11]. Of note, in clinical populations, IMUs have previously shown promise by correctly detecting foot drop recovery following lumbar spine surgery and movement abnormalities in people with cerebral palsy [12].

Machine learning has emerged in healthcare as a means of classifying large and complex datasets. Supervised models are widely used because they can predict outcomes based on labeled historical data [13,14]. Recent studies have explored the use of activity monitors combined with supervised models to classify body positions [15,16]. Previous work conducted using IMUs to measure human movement tested multiple supervised models and found that the K-Nearest Neighbours (KNN) and Support Vector Machine (SVM) models both achieved good accuracy [17].

Therefore, the overall aim of this research was to explore the use of IMUs and a machine learning approach to continuously capture, classify and visualise positions and changes in positions of people in a hospital bed.

## 2. Materials and Methods

This study was approved by the Human Research Ethics Committee at Curtin University (HREC2022-0585), and all participants provided written informed consent prior to commencing data collection. The participants were recruited via emails to academic staff at Curtin University and word of mouth. Inclusion criteria comprised the following: (i) aged ≥ 18 years, (ii) able to understand English and (iii) able to move independently between prone, supine and side lying and sustain each of these positions for ≥1 min. Potential participants were excluded if they were unable to wear an IMU on their limbs due to an amputation or their weight was greater than 180 kg, which exceeded the safe limit for the hospital bed. A target of 30 participants was set to demonstrate the applicability of this methodology. This target was based on previous studies [12,18].

Data collection took place during a single 45 min session in a dedicated teaching space located on the Curtin University campus. Within this space, there was a hospital bed, a laptop (Hewlett-Packard, Core I7 CPU, 16GB RAM, RTX 3070 Graphics Card, Perth, WA, Australia) and a RealSense camera (Intel RealSense SDK 2.0, Santa Clara, CA, USA) (Figure 1). The data captured by the RealSense camera were used for validation of the positions identified with the IMUs during data analysis.

The data collection protocol for each participant has been summarised below.

Descriptive variables (age, height, weight and gender) were recorded.Five Xsens DOT (Movella Inc., Henderson, NV, USA [19]) IMUs were placed on the participant; one on the inside side of each wrist (labelled as arm) and ankle (labelled as leg), with the fifth on the forehead (labelled as head).The camera was mounted on a tripod approximately 2 m in front of the participant’s head and connected to the Intel RealSense camera platform on a laptop. The Xsens DOTs were connected to the mobile app and initialised. The IMU sensors continuously streamed their readings via Bluetooth to the laptop, and data were recorded as comma separated value (CSV) files.The participant was instructed to assume the first position from the designated set of positions. The same instructions were given to each participant regarding positions; however, the exact angles of limbs and heads were not standardised in order to reflect natural variation between people in the same position. A 1 min interval timer was started, and camera recordings were commenced. The nine positions were chosen in consultation with expert clinicians as the most relevant positions information is required about for patients in hospital.Once the timer elapsed, recordings were ceased. The participant was instructed to maintain the current position while the camera’s rosbag and CSV file were named with the participant number, date and position number.A new recording was then initiated, and the participant was instructed to transition to the next position. Once in position, the 1 min timer was started. This process ensured that data during the transition between positions were captured as part of the recording for each specific position. All positions were captured once per participant.After each data collection session, the sensors were thoroughly cleaned before the next participant.The collected data were saved for further analysis.

The accuracy of data from Xsens DOTs has been verified in our prior work [20]. Available data were as follows:Sample time: this indicates when the reading happens.Quaternion (w, x, y, z) orientation sampled at 60 Hz: this indicates the orientation of the sensor in a three-dimensional space.Euler linear acceleration (x, y, z) sampled at 60 Hz: this indicates the acceleration of movement related to three-dimensional space.

For all positions, participants were given instructions regarding the position of their body (e.g., please lay on your back) and the position of their hands (i.e., please place your hands by your side) and asked to maintain the position for one minute. No instructions were provided regarding the position of their legs. The positions used in the protocol are shown below in Table 1.

The dataset contained both quaternion orientation and Euler linear acceleration. It was then divided into training and evaluation datasets. The evaluation dataset comprised datasets from six randomly selected participants who completed the protocol. The training dataset comprised datasets from all other participants.

Two classifiers were trained: SVM and KNN based on quaternion orientation only since the goal is to classify discrete positions rather than continuing actions. SVM is a supervised machine learning algorithm for classification and regression. The algorithm relies on three components:Hyperplane: this represents a decision boundary that separates data points into different classes.Support vectors: these are the data points closest to the decision boundary which determines the position and orientation of the hyperplane.Kernel: this is a mathematical function that maps input data into a higher-dimensional feature for classifying non-linear data.

The training process of SVM involves adjusting the support vectors to maximise the distance between the hyperplane and the closest data points. This optimisation identifies the most distinct decision boundaries for the input data classes. Once these boundaries are established, the evaluation data can be accurately classified using them.

KNN is another supervised machine learning algorithm for classification. However, instead of finding the distinct decision boundaries, KNN finds the centre points of each class where each of the data points is mapped into a high-dimensional plane. The centre points refer to points where the overall distance from all data points belonging to the corresponding class is minimised. Therefore, by calculating and comparing the distance between each centre point and the evaluation data point, the corresponding class of the closest centre point is considered as the class of the evaluation data point.

The two models were implemented with the scikit-learn Python library [https://scikit-learn.org/stable/ accessed on 22 August 2024]. Both classifiers were implemented using the default hyperparameters provided by scikit-learn. The SVM classifier utilised a Radial Basis Function (RBF) kernel, while the KNN classifier was configured with a k value of 5 and uniform weighting. Each model underwent four rounds of training and evaluation, and the accuracy was averaged across these rounds.

## 3. Results

### 3.1. Participant Details

Thirty people were recruited to this study. Participant characteristics are summarised in Table 2.

### 3.2. Machine Learning Outcomes

The accuracy of the two classifiers across all participants, trained on IMU data to correctly identify the positions, is summarised in Table 3. For each row of data, the accuracy was averaged across all positions. Confusion matrixes (CMs) were created to illustrate the accuracy of each model (Figure 2 and Figure 3) in detecting each position across four trials. Each trial had a randomly selected subset of available data from the overall dataset as the evaluation dataset. The CM captures the number of positions correctly classified, expressed as a proportion of the total number of positions assessed.

### 3.3. Visualisation of Body Position

To provide an example of how this system might be used in a hospital setting, a virtualisation system was developed. The user interface, designed in the Unity game engine (Unity, Technologies, San Francisco, CA, USA), features a generic humanoid model placed on a bed model. The joints of this humanoid were precisely synchronised with the IMU sensor readings. As shown in Figure 4, the user interface provides a real-time visualisation of the person’s position and issues alerts when suboptimal positioning or prolonged immobility is detected.

Additionally, in Figure 5, an example dashboard within the user interface is shown. This could be used by healthcare practitioners to explore summary data regarding a patient’s position over a longer period of time and provide prompts for the healthcare practitioner to consider changing the patient’s position.

## 4. Discussion

To the best of our knowledge, this is the first study to explore whether IMUs together with a machine learning approach can, in real time, accurately detect and display the body position of people in a hospital-like setting. The important findings of this study are as follows: (i) data from the Xsens DOTs with either the SVM or KNN machine learning approach, when averaged over four trials, accurately classified the position of a person in a single hospital bed ≥85% of the time; (ii) with both machine learning approaches, the combination of sensors that provided the most accurate results was left and right wrist sensors together with a forehead sensor; (iii) including data from the ankle sensors reduced the accuracy of both machine learning approaches; and (iv) a user interface, developed using the Unity game engine, allowed real-time visualisation of current positions and a record of time previously spent in positions.

Using data from the left and right wrist sensors and the forehead sensor, the highest average accuracies over four trials for all positions were similar for the SVM and KNN machine learning approaches (87.7 vs. 85.8%, respectively). Notably, this is consistent with an earlier study showing that wearables and the SVM classifier could predict ground reaction forces in ballet dancers during unilateral and bilateral jumps with 87.8% and 80.8% accuracy, respectively [21]. Despite the similarity between the SVM and KNN machine learning approaches over four trials for all positions, when data from individual positions were explored in the CMs, SVM classified nine positions with 74% to 100% accuracy across all trials except trial 3. In contrast, KNN only achieved 43% accuracy in trial 2, which was 30% lower than SVM in the same trial. The indicates the superiority of the SVM machine learning approach in this context. Of note, both models were able to classify standing bedside the bed (position I) with reasonable accuracy. This is likely because the IMU data only represent orientation, and position I is the same as position A with two 90-degree rotations along two axes.

For both machine learning approaches, perhaps surprisingly, performance did not peak when the maximum number of sensors was used. Indeed, adding data from the ankle sensors appeared to introduce noise and reduced the accuracy of both SVM and KNN. For both machine learning approaches, the highest accuracy was achieved when sensors were placed on the wrists and forehead, suggesting that data from these sensors, compared with data from ankle sensors, were more specific to and consistent with individual positions. Specifically, in many instances, it was the wrist position only that differentiated positions (e.g., supine, hands by side vs. supine, hands under head). Indeed, using wrist sensors without any other sensor resulted in reasonable accuracy (83.5% with SVM and 81.7% with KNN). Using wrist sensors alone with the SVM machine learning approach would be a pragmatic choice in the hospital setting as applying less sensors would reduce the burden on clinical staff with minimal trade-off in the overall accuracy.

This study adds to the growing body of literature examining the use of sensors to explore body position in a healthcare setting. Kroll et al. (2020) [22] conducted a feasibility study using a combination of visual and acoustic sensors and a sensor mat under pressure to capture bodily movement. The system was able to capture movement but not classify positions. A small study (n = 6) by Casas et al. (2019) [23] used a movement sensor system to classify positions in a simulated clinical environment. Instead of wearable sensors, the system used a pressure-sensing mattress and deep learning model. The model was able to accurately estimate five different positions with an overall joint position error of 8 cm and a limb orientation error of 19 degrees. One limitation noted in this study was that the system performed poorly when limbs were not in contact with the pressure mattress. In contrast, our data demonstrate that IMUs together with the SVM machine learning approach could accurately classify positions and changes in position, including positions when the person moved out of the bed, and these data can be presented in a usable way when put through a user interface.

Although the data presented in the current study support the accuracy of using left and right wrist sensors with a forehead sensor with an SVM classifier to classify the body position of adults in a hospital bed, the implementation of such technology in hospitals for people who are unwell is likely to be challenging. A recent review summarised the current literature regarding the use of machine learning in intensive care units and highlighted that the general public may not accept or trust such systems to drive healthcare decisions [24]. The public are also likely to raise concerns regarding security breaches, maintenance of privacy and anonymity and want safeguards around the misuse of data [24]. These factors will need careful consideration before such technology can be implemented in a hospital setting. Studies that partner with consumers to authentically understand and address these concerns are needed.

A major strength of this study was the interdisciplinary team of investigators, with expertise from both engineering and physiotherapy. We were also careful to conduct data collection in a simulation suite that closely represented a ward hospital bed. The study sample varied in age, height and weight and was broadly representative of the demographics of people admitted to a hospital. The Xsens DOTs were small and unobtrusive and were well tolerated by the study participants.

The main limitation of this study was that we did not explore all possible positions that people may adopt in a hospital bed. Further, the scope of this study was limited to exploring positions and changes in positions, and further work is needed to explore parameters that describe the movement of individual body parts and the speed and duration of the position change. Finally, the battery life of the Xsens DOT is unlikely to extend beyond 6 h and limits the capacity of this system to be used for prolonged continuous monitoring (e.g., 24 h periods).

## 5. Conclusions

Sensors attached to both wrists, together with an SVM machine learning program, were able to accurately identify 83.5% of positions. Adding data from a forehead sensor increased the accuracy to 87.7%. Further work is needed to (i) extend this methodology to allow accurate measurements of the movement of individual body parts (i.e., quantity, symmetry, quality, etc.), (ii) explore ways to integrate measures of position and movement with physiological data and assess the sensitivity of the system to detect early deterioration in clinical status, (iii) explore factors that will facilitate the acceptance of this system in a live clinical environment by patients and healthcare providers and (iv) validate the system in a clinical environment.

## Figures and Tables

**Figure 1 sensors-25-00499-f001:**
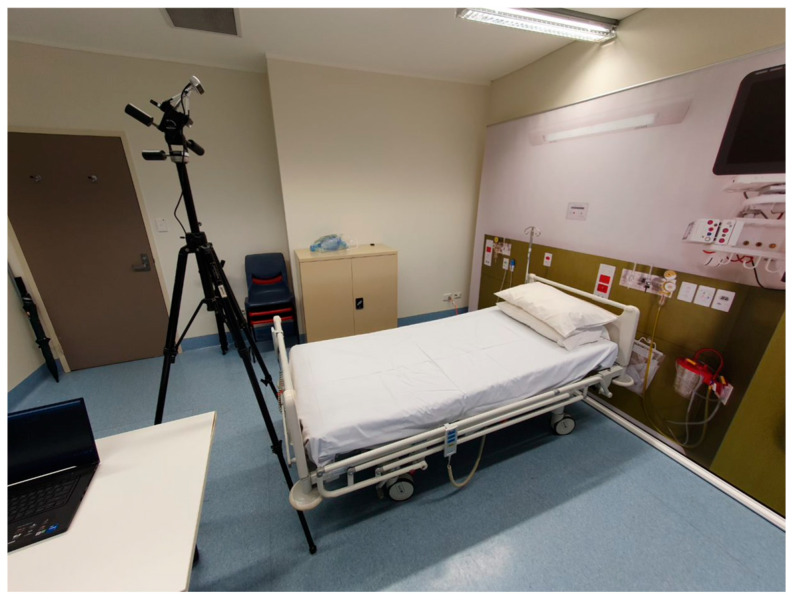
Laboratory setup used for data collection.

**Figure 2 sensors-25-00499-f002:**
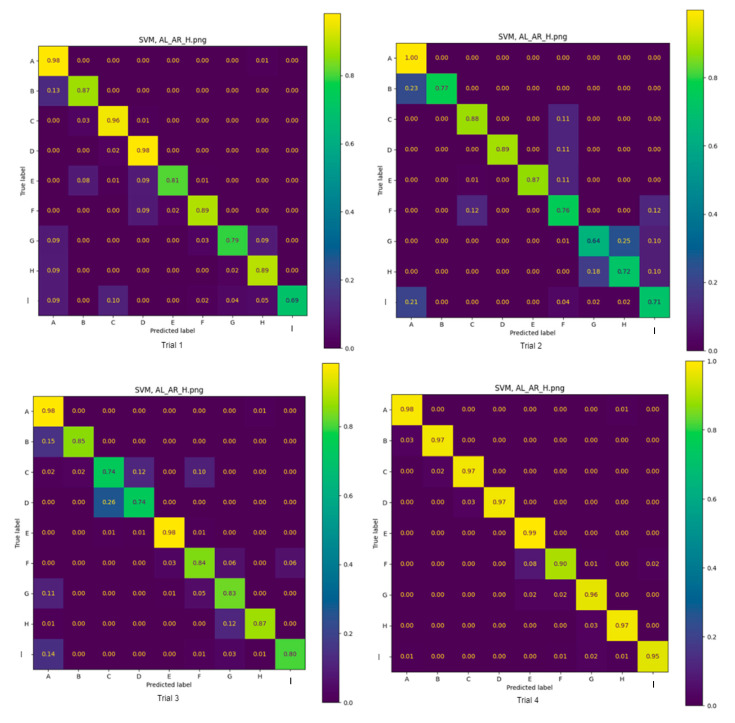
Four trials: SVM AL_AR_H confusion matrix.

**Figure 3 sensors-25-00499-f003:**
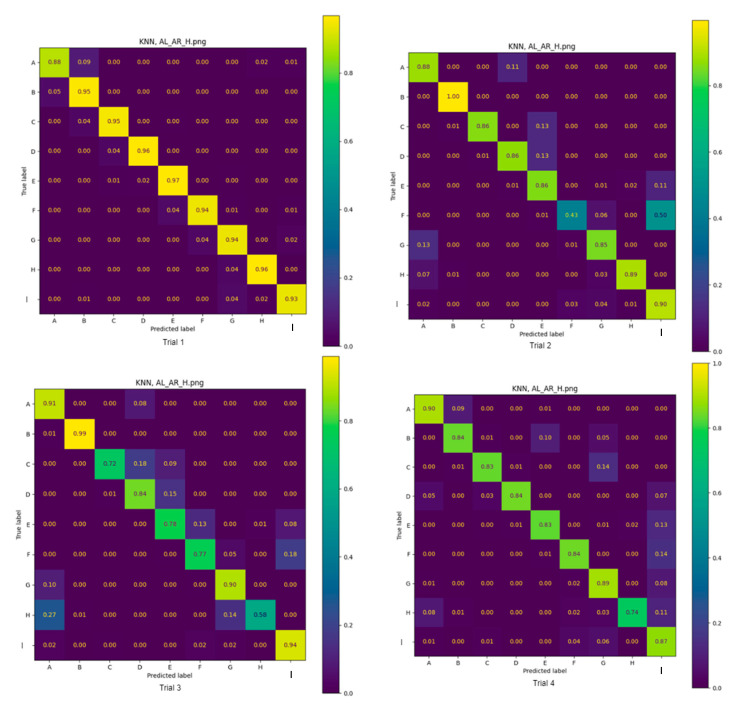
Four trials: KNN AL_AR_H confusion matrix.

**Figure 4 sensors-25-00499-f004:**
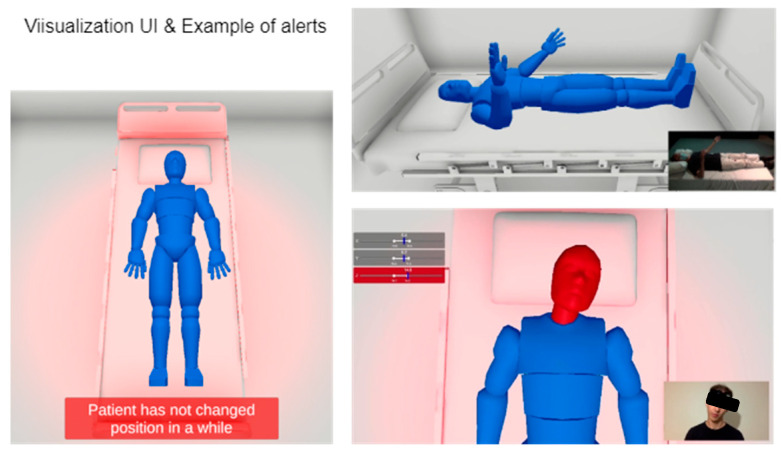
Visualisation of user interface including alert system (shown in red).

**Figure 5 sensors-25-00499-f005:**
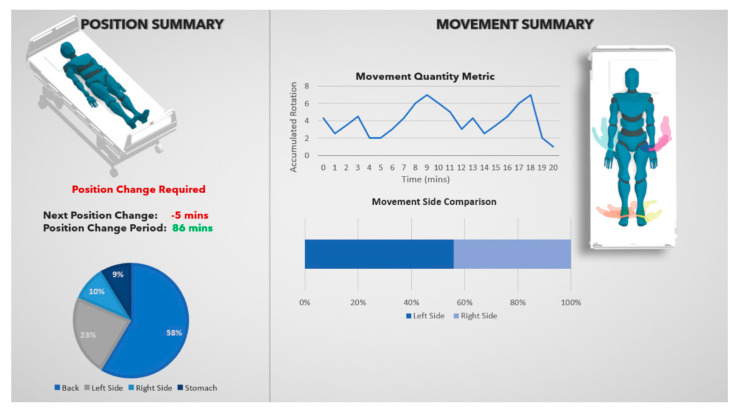
Dashboard user interface.

**Table 1 sensors-25-00499-t001:** Positions adopted during data collection.

No.	Position of Body	Position of Hands	Label
1	Supine	Hands by side	A
2	Supine	Hands under head	B
3	Prone	Hands by side	E
4	Prone	Hands under head	F
5	Left-side lying	Hands under head	C
6	Left-side lying	left hand on abdomen	D
7	left-side lying	Hands under head	G
8	left-side lying	Left hand on abdomen	H
9	Standing beside bed	Hands by side	I

**Table 2 sensors-25-00499-t002:** Participant characteristics (*n* = 30).

Variable	Mean (SD)
Age	45 years (19 years)
Gender	15 Female
Height	175 cm (8 cm)
Weight	74 kg (14 kg)

**Table 3 sensors-25-00499-t003:** Highest average accuracy for each number of sensors and their corresponding combination of sensors for SVM and KNN classifiers.

Number of Sensors	SVM	Accuracy	KNN	Accuracy
1	AR	0.6465	AR	0.5693
2	AL_AR	0.8351	AL_AR	0.8176
3	AL_AR_H	0.8773	AL_AR_H	0.8583
4	AL_AR_LL_H	0.8499	AL_AR_LL_LR	0.7780
5	AL_AR_LL_LR_H	0.6661	AL_AR_LL_LR_H	0.7028

Definitions of abbreviations: AL: arm left; AR: arm right; LL: leg left; LR: leg right; and H: head.

## Data Availability

The raw data supporting the conclusions of this article will be made available by the authors on request.

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
