# Peer review of "Using Inertial Measurement Units and Machine Learning to Classify Body Positions of Adults in a Hospital Bed"

_sensors, 2025, doi:10.3390/s25020499_

Round 1

Reviewer 1 Report

Comments and Suggestions for Authors

Your study investigates using IMUs and machine learning to track and visualize patient body positions in hospital beds.Participants assumed various positions while wearing IMU sensors, and their movements were analyzed using SVM and KNN algorithms.Forehead and wrist sensors provided the most accurate results, and a Unity-based interface was developed for real-time monitoring and alerts.

The topic is interesting, but there are some major concerns that that need to be addressed:

  • Small Sample Size: The study recruited only 30 participants, which may not fully represent the diverse body position characteristics of all hospital patients, necessitating further validation with larger sample sizes.

  • Limited Position Scope: The study focused on only ten common body positions, whereas patients may adopt a wider variety of positions, requiring further research on more comprehensive position classification methods.

  • Omission of Movement Dynamics: Merely identifying body positions is insufficient; it is crucial to consider the speed and duration of position changes to assess potential danger to the patient.

  • Lack of Clinical Application: The study was conducted solely in a simulated environment, necessitating further validation in actual clinical settings to ensure system reliability and practicality.

  • Privacy Concerns: The study does not address patient privacy considerations, such as data encryption or anonymization techniques, which need to be addressed before implementation.

  • It is recommended to cite the following articles to improve the content of the paper and make it more comprehensive:

â‘ T. R. N and R. Gupta, "A Survey on Machine Learning Approaches and Its Techniques:," 2020 IEEE International Students' Conference on Electrical,Electronics and Computer Science (SCEECS), Bhopal, India, 2020, pp. 1-6, doi: 10.1109/SCEECS48394.2020.190.

â‘¡T. R. N and R. Gupta, "Feature Selection Techniques and its Importance in Machine Learning: A Survey," 2020 IEEE International Students' Conference on Electrical,Electronics and Computer Science (SCEECS), Bhopal, India, 2020, pp. 1-6, doi: 10.1109/SCEECS48394.2020.189.

â‘¢Fo Hu, Mengyuan Qian, Kailun He, Wen-an Zhang, Xusheng Yang. A Novel Multi-Feature Fusion Network with Spatial Partitioning Strategy and Cross-Attention for Armband-Based Gesture Recognition[J]. IEEE Transactions on Neural Systems and Rehabilitation Engineering. 2024, 32: 3878-3890.

â‘£Hu F, Zhang L, Yang X, et al. EEG-Based Driver Fatigue Detection Using Spatio-Temporal Fusion Network With Brain Region Partitioning Strategy[J]. IEEE Transactions on Intelligent Transportation Systems, 2024, 25(8): 9618-9630.

⑤T. Qin, P. Li and S. Shen, "VINS-Mono: A Robust and Versatile Monocular Visual-Inertial State Estimator," in IEEE Transactions on Robotics, vol. 34, no. 4, pp. 1004-1020, Aug. 2018, doi: 10.1109/TRO.2018.2853729.

â‘¥K. Qiu, T. Qin, W. Gao and S. Shen, "Tracking 3-D Motion of Dynamic Objects Using Monocular Visual-Inertial Sensing," in IEEE Transactions on Robotics, vol. 35, no. 4, pp. 799-816, Aug. 2019, doi: 10.1109/TRO.2019.2909085. 

Comments on the Quality of English Language
  • Pay attention to the use of articles: In some instances, the use of articles can be made more precise. For example, “a hospital bed” could be revised to “a single hospital bed,” and “a Unity game engine” could be changed to “the Unity game engine.”

  • Avoid repetition: The article contains instances of repetitive words or phrases, such as “body position” and “body posture.” Consider employing synonyms or pronouns to avoid repetition.

  • Use more precise verbs: In certain cases, more specific verbs can be used to describe actions. For instance, “started” could be replaced with “initiated,” and “stopped” could be changed to “ceased.”

Reviewer 2 Report

Comments and Suggestions for Authors

Thanks for the article submission, but I'm afraid this paper has some serious flaws. 

1. The introduction talks a lot about deterioration of health and vital signs and pressure ulcer prevention is almost an afterthought in lines 56-59. Yet, the results section with figures 4 and 5 clearly shows that you were mostly interested in pressure ulcer applications. The introduction should match the results/discussion.

2. Regarding the introduction, you say in lines 61-63 that "multiple methods have been developed that allow continuous monitoring of bodily position and movement, such as use of optical motion capture technologies, Fiber-optic sensors, E-textile sensors and Inertial Measurement Units (IMU)." However, you omitted using force plates/load cells such as these two references which are actually very similar to what you are trying to accomplish, just with a different sensor. I'd suggest focusing the introduction more on pressure ulcers and include the load cell method.

Duvall J, Karg P, Brienza D, Pearlman J. Detection and classification methodology for movements in the bed that supports continuous pressure injury risk assessment and repositioning compliance. J Tissue Viability. 2019 Feb;28(1):7-13. doi: 10.1016/j.jtv.2018.12.001. Epub 2018 Dec 24. PMID: 30598376; PMCID: PMC6382541.

Pupic, N.; Gabison, S.; Evans, G.; Fernie, G.; Dolatabadi, E.; Dutta, T. Detecting Patient Position Using Bed-Reaction Forces for Pressure Injury Prevention and Management. Sensors 202424, 6483. https://doi.org/10.3390/s24196483

3. In Lines 112-115, you say "A new recording was then initiated, and the participant was instructed to transition to the next position. Once in position, the 1-minute timer was started. This process ensured data during the transition between positions were captured as part of the recording for each specific position." This makes no sense to me. Why would data between positions be included in the recordings/data for each position? If you're classifying positions, how they arrived at a position shouldn't matter and will only confuse the classifier.

4. What data was actually used in the classifiers? Is it the three things listed in lines 122-126? If so, why use sample time or acceleration? You would essentially just want the orientation of the sensors to classify position. Also, what values were used? Mean, peak, standard deviations? You also don't mention sampling frequencies for the IMUs or the Realsense camera.

5. What was the realsense camera used for? I don't believe it was involved in the classification.

6. You need to state how many trials were completed and hence the number of data points and variables used. You only state the number of participants and positions. For example, if only one trial was completed then you would have 24 (participants) x 9 (positions) data sets in the training set to evaluate 6 (participants) x 9 (positions) data sets. 

Here is another list of things that don't make sense or need explaining:

1. The confusion matrices don't make sense with the accuracies. If the SVM for AL_AR_H confusion matrix in Figure 4 has a minimum accuracy for each position above .90, how is the overall accuracy in Table 3 0.8773? Assuming you had equal numbers for each position, the overall accuracy should be the mean of the diagonal numbers on the confusion matrix.

2. I would really like an explanation of how the I position (standing next to the bed) possibly had an error in classification considering every orientation value should have been vastly different from all other positions (being horizontal on the bed). 

3. On lines 246-248 you state "Specifically, in many instances, it was wrist position only that differentiated positions (e.g. supine, hands by side vs supine, hands under head). This makes sense because these are the positions you chose. To differentiate the two supine, two prone, two left side, and two right side positions, the only difference was hand/arm placement so naturally the wrist sensors would be the ones that would differentiate those positions. The question I would ask is why does hand placement matter as it relates to pressure ulcer prevention? If someone was in a bed laying supine and moving back and forth between positions A and B, that still means they are having pressure on their sacrum and at risk for an ulcer.

Round 2

Reviewer 1 Report

Comments and Suggestions for Authors

N/A

Comments on the Quality of English Language

N/A

Author Response

There are no attached files. 

Reviewer 2 Report

Comments and Suggestions for Authors

Thanks for the revision. It was very helpful, and the paper is much improved. You addressed most of my comments, but I have a couple more questions.

1. When you say trials in section 3.2 and Figure 4 is that the same as participants since you said each participant did 1 trial of each position? If so, please say participant rather than trial.

2. I think a visual representation or actual images of a researcher in the positions would help as I'm still very confused as to the misclassifications of the classifier especially between position I and A. After thinking about it more, one question I have is how much control was there in positions as this could be a big factor. For example, which way were they facing when standing beside the bed in position I (towards the bed, away from the bed, facing the foot or facing the head) or was that not controlled? Also, in position A and C with hands by the sides, were their hands palm down/up or palms toward the body or was that not controlled. For the prone positions (C and D), I'm assuming their heads were turned one way or another rather than face towards the bed. Was that controlled? If these weren't controlled, I can more understand discrepancies since the same positions could have some axes 90 or even 180 degrees different. For example, if one participant in position C turned their head to left and another turned their head to the right. The head sensor would be 180 degrees different for the same position.  

Either way, I think some discussion of how much control was placed on these positions is warranted in the discussion section. I also think a statement in the methods section that each IMU was oriented the same way for each participant is needed.
